# Improving Online Algorithms via ML Predictions

**Ravi Kumar**
Google
Mountain View, CA
ravi.k53@gmail.com

**Manish Purohit**
Google
Mountain View, CA
mpurohit@google.com

**Zoya Svitkina**
Google
Mountain View, CA
zoya@cs.cornell.edu

## Abstract

In this work we study the problem of using machine-learned predictions to improve the performance of online algorithms. We consider two classical problems, ski rental and non-clairvoyant job scheduling, and obtain new online algorithms that use predictions to make their decisions. These algorithms are oblivious to the performance of the predictor, improve with better predictions, but do not degrade much if the predictions are poor.

## 1   Introduction

Dealing with uncertainty is one of the most challenging issues that real-world computational tasks, besides humans, face. Ranging from "will it snow next week?" to "should I rent an apartment or buy a house?", there are questions that cannot be answered reliably without some knowledge of the future. Similarly, the question of "which job should I run next?" is hard for a CPU scheduler that does not know how long this job will run and what other jobs might arrive in the future.

There are two interesting and well-studied computational paradigms aimed at tackling uncertainty. The first is in the field of machine learning where uncertainty is addressed by making predictions about the future. This is typically achieved by examining the past and building robust models based on the data. These models are then used to make predictions about the future. Humans and real-world applications can use these predictions to adapt their behavior: knowing that it is likely to snow next week can be used to plan a ski trip. The second is in the field of algorithm design. Here, the effort has to been to develop a notion of competitive ratio[1] for the goodness of an algorithm in the presence of an unknown future and develop online algorithms that make decisions heedless of the future but are provably good in the *worst-case*, i.e., even in the most pessimistic future scenario. Such online algorithms are popular and successful in real-world systems and have been used to model problems including paging, caching, job scheduling, and more (see the book by Borodin and El-Yaniv [5]).

Recently, there has been some interest in using machine-learned predictions to improve the quality of online algorithms [20, 18]. The main motivation for this line of research is two-fold. The first is to design new online algorithms that can avoid assuming a worst-case scenario and hence have better performance guarantees both in theory and practice. The second is to leverage the vast amount of modeling work in machine learning, which precisely deals with how to make predictions. Furthermore, as machine-learning models are often retrained on new data, these algorithms can naturally adapt to evolving data characteristics. When using the predictions, it is important that the online algorithm is unaware of the performance of the predictor and makes no assumptions on the types of prediction errors. Additionally, we desire two key properties of the algorithm: (i) if the predictor is good, then the online algorithm should perform close to the best offline algorithm (*consistency*) and (ii) if the predictor is bad, then the online algorithm should gracefully degrade, i.e., its performance should be close to that of the online algorithm without predictions (*robustness*).

**Our problems.** We consider two basic problems in online algorithms and show how to use machine-learned predictions to improve their performance in a provable manner. The first is *ski rental*, in which a skier is going to ski for an unknown number of days and on each day can either rent skis at unit price or buy them for a higher price $b$ and ski for free from then on. The uncertainty is in the number of skiing days, which a predictor can estimate. Such a prediction can be made reasonably well, for example, by building models based on weather forecasts and past behavior of other skiers. The ski rental problem is the canonical example of a large class of online rent-or-buy problems, which arise whenever one needs to decide between a cheap short-term solution ("renting") and an expensive long-term one ("buying"). Several extensions and generalizations of the ski rental problem have been studied leading to numerous applications such as dynamic TCP acknowledgement [11], buying parking permits [21], renting cloud servers [14], snoopy caching [13], and others. The best known deterministic algorithm for ski rental is the *break-even* algorithm: rent for the first $b - 1$ days and buy on day $b$. It is easy to observe that the break-even algorithm has a competitive ratio of 2 and no deterministic algorithm can do better. On the other hand, Karlin et al. [12] designed a randomized algorithm that yields a competitive ratio of $\frac{e}{e-1} \approx 1.58$, which is also optimal.

The second problem we consider is *non-clairvoyant job scheduling*. In this problem a set of jobs, all of which are available immediately, have to be scheduled on one machine; any job can be preempted and resumed later. The objective is to minimize the sum of completion times of the jobs. The uncertainty in this problem is that the scheduler does not know the running time of a job until it actually finishes. Note that a predictor in this case can predict the running time of a job, once again, by building a model based on the characteristics of the job, resource requirements, and its past behavior. Non-clairvoyant job scheduling, introduced by Motwani et al. [23], is a basic problem in online algorithms with a rich history and, in addition to its obvious applications to real-world systems, many variants and extensions of it have been studied extensively in the literature [9, 3, 1, 10]. Motwani et al. [23] showed that the round-robin algorithm has a competitive ratio of 2, which is optimal.

**Main results.** Before we present our main results we need a few formal notions. In online algorithms, the *competitive ratio* of an algorithm is defined as the worst-case ratio of the algorithm cost to the offline optimum. In our setting, this is a function $c(\eta)$ of the error $\eta$ of the predictor[2]. We say that an algorithm is $\gamma$-*robust* if $c(\eta) \leq \gamma$ for all $\eta$, and that it is $\beta$-*consistent* if $c(0) = \beta$. So consistency is a measure of how well the algorithm does in the best case of perfect predictions, and robustness is a measure of how well it does in the worst-case of terrible predictions.

Let $\lambda \in (0, 1)$ be a hyperparameter. For the ski rental problem with a predictor, we first obtain a deterministic online algorithm that is $(1 + 1/\lambda)$-robust and $(1 + \lambda)$-consistent (Section 2.2). We next improve these bounds by obtaining a randomized algorithm that is $(\frac{1}{1 - e^{-(\lambda - 1/b)}})$-robust and $(\frac{\lambda}{1 - e^{-\lambda}})$-consistent, where $b$ is the cost of buying (Section 2.3). For the non-clairvoyant scheduling problem, we obtain a randomized algorithm that is $(2/(1 - \lambda))$-robust and $(1/\lambda)$-consistent. Note that the consistency bounds for all these algorithms circumvent the lower bounds, which is possible only because of the predictions.

It turns out that for these problems, one has to be careful how the predictions are used. We illustrate through an example that if the predictions are used naively, one cannot ensure robustness (Section 2.1). Our algorithms proceed by opening up the classical online algorithms for these problems and using the predictions in a judicious manner. We also conduct experiments to show that the algorithms we develop are practical and achieve good performance compared to ones that do not use any prediction.

**Related work.** The work closest to ours is that of Medina and Vassilvitskii [20] and Lykouris and Vassilvitskii [18]. The former used a prediction oracle to improve reserve price optimization, relating the gap beween the expected bid and revenue to the average predictor loss. In a sense, this paper initiated the study of online algorithms equipped with machine learned predictions. The latter developed this framework further, introduced the concepts of robustness and consistency, and considered the online caching problem with predictions. It modified the well-known Marker algorithm to use the predictions ensuring both robustness and consistency. While we operate in the same framework, none of their techniques are applicable to our setting. Another recent work is that of Kraska et al. [17] that empirically shows that better indexes can be built using machine learned models; it does not provide any provable guarantees for its methods.

There are other computational models that try to tackle uncertainty. The field of robust optimization [16] considers uncertain inputs and aims to design algorithms that yield good performance guarantees for any potential realization of the inputs. There has been some work on analyzing algorithms when the inputs are stochastic or come from a known distribution [19, 22, 6]. In the optimization community, the whole field of online stochastic optimization concerns online decision making under uncertainty by assuming a distribution on future inputs; see the book by Russell Bent and Pascal Van Hentenryck [4]. Our work differs from these in that we do not assume anything about the input; in fact, we do not assume anything about the predictor either!

## 2 Ski rental with prediction

In the ski rental problem, let rentals cost one unit per day, $b$ be the cost to buy, $x$ be the actual number of skiing days, which is unknown to the algorithm, and $y$ be the predicted number of days. Then $\eta = |y - x|$ is the prediction error. Note that we do not make any assumptions about its distribution. The optimum cost is $\mathsf{OPT} = \min\{b, x\}$.

### 2.1 Warmup: A simple consistent, non-robust algorithm

We first show that an algorithm that naively uses the predicted number of days to decide whether or not to buy is 1-consistent, i.e., its competitive ratio is 1 when $\eta = 0$. However, this algorithm is not robust, as the competitive ratio can be arbitrarily large in case of incorrect predictions.

---
**Algorithm 1:** A simple 1-consistent algorithm

**if** $y \geq b$ **then**
    Buy on the first day.
**else**
    Keep renting for all skiing days.
**end**

---

**Lemma 2.1.** *Let* $\mathsf{ALG}$ *denote the cost of the solution obtained by Algorithm 1 and let* $\mathsf{OPT}$ *denote the optimal solution cost on the same instance. Then* $\mathsf{ALG} \leq \mathsf{OPT} + \eta$.

*Proof.* We consider different cases based on the relative values of the prediction $y$ and the actual number of days $x$ of the instance. Recall that Algorithm 1 incurs a cost of $b$ whenever the prediction is at least $b$ and incurs a cost of $x$ otherwise.

- $y \geq b, x \geq b \implies \mathsf{ALG} = b = \mathsf{OPT}$.
- $y < b, x < b \implies \mathsf{ALG} = x = \mathsf{OPT}$
- $y \geq b, x < b \implies \mathsf{ALG} = b \leq x + y - x = x + \eta = \mathsf{OPT} + \eta$
- $y < b, x \geq b \implies \mathsf{ALG} = x < b + x - y = b + \eta = \mathsf{OPT} + \eta$     □

A major drawback of Algorithm 1 is its lack of robustness. In particular, its competitive ratio can be unbounded if the prediction $y$ is small but $x \gg b$. Our goal next is to obtain an algorithm that is both consistent and robust.

### 2.2 A deterministic robust and consistent algorithm

In this section, we show that a small modification to Algorithm 1 yields an algorithm that is both consistent and robust. Let $\lambda \in (0, 1)$ be a hyperparameter. As we see later, varying $\lambda$ gives us a smooth trade-off between the robustness and consistency of the algorithm.

**Theorem 2.2.** *With a parameter* $\lambda \in (0, 1)$*, Algorithm 2 has a competitive ratio of at most* $\min\left\{\dfrac{1+\lambda}{\lambda}, (1+\lambda) + \dfrac{\eta}{(1-\lambda)\mathsf{OPT}}\right\}$*. In particular, Algorithm 2 is* $(1 + 1/\lambda)$*-robust and* $(1+\lambda)$*-consistent.*

*Proof.* We begin with the first bound. Suppose $y \geq b$ and the algorithm buys the skis at the start of day $\lceil \lambda b \rceil$. Since the algorithm incurs a cost of $b + \lceil \lambda b \rceil - 1$ whenever $x \geq \lceil \lambda b \rceil$, the worst competitive

---
**Algorithm 2:** A deterministic robust and consistent algorithm.

> **if** $y \geq b$ **then**
>> Buy on the start of day $\lceil \lambda b \rceil$
>
> **else**
>> Buy on the start of day $\lceil b/\lambda \rceil$
>
> **end**
---

ratio is obtained when $x = \lceil \lambda b \rceil$, for which $\mathsf{OPT} = \lceil \lambda b \rceil$. In this case, we have $\mathsf{ALG} = b + \lceil \lambda b \rceil - 1 \leq b + \lambda b \leq \left(\frac{1+\lambda}{\lambda}\right) \lceil \lambda b \rceil = \left(\frac{1+\lambda}{\lambda}\right) \mathsf{OPT}$. On the other hand, when $y < b$, the algorithm buys skis at the start of day $\lceil b/\lambda \rceil$ and rents until then. In this case, the worst competitive ratio is attained whenever $x = \lceil b/\lambda \rceil$ as we have $\mathsf{OPT} = b$ and $\mathsf{ALG} = b + \lceil b/\lambda \rceil - 1 \leq b + b/\lambda = \left(\frac{1+\lambda}{\lambda}\right) \mathsf{OPT}$.

To prove the second bound, we need to consider the following two cases. Suppose $y \geq b$. Then, for all $x < \lceil \lambda b \rceil$, we have $\mathsf{ALG} = \mathsf{OPT} = x$. On the other hand, for $x \geq \lceil \lambda b \rceil$, we have $\mathsf{ALG} = b + \lceil \lambda b \rceil - 1 \leq (1 + \lambda)b \leq (1 + \lambda)(\mathsf{OPT} + \eta)$. The second inequality follows since either $OPT = b$ (if $x \geq b$) or $b \leq y \leq \mathsf{OPT} + \eta$ (if $x < b$). Suppose $y < b$. Then, for all $x \leq b$, we have $\mathsf{ALG} = \mathsf{OPT} = x$. Similarly, for all $x \in (b, \lceil b/\lambda \rceil)$, we have $\mathsf{ALG} = x \leq y + \eta < b + \eta = \mathsf{OPT} + \eta$. Finally for all $x \geq \lceil b/\lambda \rceil$, noting that $\eta = x - y > b/\lambda - b = (1 - \lambda)b/\lambda$, we have $\mathsf{ALG} = b + \lceil b/\lambda \rceil - 1 \leq b + b/\lambda < b + (\frac{1}{1-\lambda})\eta = \mathsf{OPT} + (\frac{1}{1-\lambda})\eta$. Thus we obtain $\mathsf{ALG} \leq (1 + \lambda)\mathsf{OPT} + (\frac{1}{1-\lambda})\eta$, completing the proof. $\qquad\square$

Thus, Algorithm 2 gives an option to trade-off consistency and robustness. In particular, greater trust in the predictor suggests setting $\lambda$ close to zero as this leads to a better competitive ratio when $\eta$ is small. On the other hand, setting $\lambda$ close to one is conservative and yields a more robust algorithm.

### 2.3 A randomized robust and consistent algorithm

In this section we consider a family of randomized algorithms and compare their performance against an oblivious adversary. In particular, we design robust and consistent algorithms that yield a better trade-off than the above deterministic algorithms. Let $\lambda \in (1/b, 1)$ be a hyperparameter. For a given $\lambda$, Algorithm 3 samples the day when skis are bought based on two different probability distributions, depending on the prediction received, and rents until that day.

---
**Algorithm 3:** A randomized robust and consistent algorithm

> **if** $y \geq b$ **then**
>> Let $k \leftarrow \lfloor \lambda b \rfloor$;
>>
>> Define $q_i \leftarrow \left(\frac{b-1}{b}\right)^{k-i} \cdot \frac{1}{b(1-(1-1/b)^k)}$ for all $1 \leq i \leq k$;
>>
>> Choose $j \in \{1 \ldots k\}$ randomly from the distribution defined by $q_i$;
>>
>> Buy at the start of day $j$.
>
> **else**
>> Let $\ell \leftarrow \lceil b/\lambda \rceil$;
>>
>> Define $r_i \leftarrow \left(\frac{b-1}{b}\right)^{\ell-i} \cdot \frac{1}{b(1-(1-1/b)^\ell)}$ for all $1 \leq i \leq \ell$;
>>
>> Choose $j \in \{1 \ldots \ell\}$ randomly from the distribution defined by $r_i$;
>>
>> Buy at the start of day $j$.
>
> **end**
---

**Theorem 2.3.** *Algorithm 3 yields a competitive ratio of at most* $\min\{\frac{1}{1-e^{-(\lambda-1/b)}}, \frac{\lambda}{1-e^{-\lambda}}(1 + \frac{\eta}{\mathsf{OPT}})\}$. *In particular, Algorithm 3 is* $\left(\frac{1}{1-e^{-(\lambda-1/b)}}\right)$-*robust and* $\left(\frac{\lambda}{1-e^{-\lambda}}\right)$-*consistent.*

*Proof.* We consider different cases depending on the relative values of $y$ and $x$.

(i) $y \geq b, x \geq k$. Here, we have $\mathsf{OPT} = \min\{b, x\}$. Since the algorithm incurs a cost of $(b + i - 1)$ when we buy at the beginning of day $i$, we have

$$\mathbb{E}[\mathsf{ALG}] = \sum_{i=1}^{k}(b + i - 1)q_i = \sum_{i=1}^{k}(b + i - 1)\left(\frac{b-1}{b}\right)^{k-i}\frac{1}{b(1-(1-1/b)^k)} = \frac{k}{1-(1-1/b)^k}$$

$$\leq \frac{k}{1 - e^{-k/b}} \;\leq\; \left( \frac{k/b}{1 - e^{-k/b}} \right) (\mathsf{OPT} + \eta) \;\leq\; \left( \frac{\lambda}{1 - e^{-\lambda}} \right) (\mathsf{OPT} + \eta).$$

(ii) $y \geq b, x < k$. Here, we have $\mathsf{OPT} = x$. On the other hand, the algorithm incurs a cost of $(b + i - 1)$ only if it buys at the beginning of day $i \leq x$. In particular, we have

$$\mathbb{E}[\mathsf{ALG}] = \sum_{i=1}^{x} (b + i - 1) q_i + \sum_{i=x+1}^{k} x q_i$$

$$= \frac{1}{b(1 - (1 - 1/b)^k)} \left[ \sum_{i=1}^{x} (b + i - 1) \left( \frac{b-1}{b} \right)^{k-i} + \sum_{i=x+1}^{k} x \left( \frac{b-1}{b} \right)^{k-i} \right]$$

$$= \frac{x}{1 - (1 - 1/b)^k} \;\leq\; \left( \frac{1}{1 - e^{-k/b}} \right) \mathsf{OPT} \;\leq\; \left( \frac{1}{1 - e^{-(\lambda - 1/b)}} \right) \mathsf{OPT},$$

which establishes robustness. In order to prove consistency, we can rewrite the RHS as follows

$$\mathbb{E}[\mathsf{ALG}] \leq \left( \frac{1}{1 - e^{-k/b}} \right) \mathsf{OPT} \;=\; \left( \frac{k/b}{1 - e^{-k/b}} \right) \mathsf{OPT} + \left( \frac{(b-k)/b}{1 - e^{-k/b}} \right) x$$

$$\leq \left( \frac{k/b}{1 - e^{-k/b}} \right) \mathsf{OPT} + \left( \frac{k/b}{1 - e^{-k/b}} \right) \eta \;\leq\; \left( \frac{\lambda}{1 - e^{-\lambda}} \right) (\mathsf{OPT} + \eta),$$

since $x < k$ and $b - k \leq \eta$.

(iii) $y < b, x < \ell$. Here, we have $\mathsf{OPT} = \min\{b, x\}$. On the other hand, the expected cost of the algorithm can be computed similar to (ii)

$$\mathbb{E}[\mathsf{ALG}] = \sum_{i=1}^{x} (b + i - 1) r_i + \sum_{i=x+1}^{\ell} x r_i \;\leq\; \left( \frac{1}{1 - e^{-\ell/b}} \right) x$$

$$\leq \left( \frac{1}{1 - e^{-1/\lambda}} \right) (\mathsf{OPT} + \eta) \;\leq\; \left( \frac{\lambda}{1 - e^{-\lambda}} \right) (\mathsf{OPT} + \eta).$$

(iv) $y < b, x \geq \ell$. Here, we have $\mathsf{OPT} = b$. The expected cost incurred by the algorithm is as in (i).

$$\mathbb{E}[\mathsf{ALG}] = \sum_{i=1}^{\ell} (b + i - 1) r_i \;=\; \frac{\ell}{1 - (1 - 1/b)^{\ell}} \;\leq\; \frac{\lceil b/\lambda \rceil}{(1 - e^{-\ell/b})}$$

$$\leq \left( \frac{1/\lambda + 1/b}{(1 - e^{-1/\lambda})} \right) \mathsf{OPT} \;\leq\; \left( \frac{1}{1 - e^{-(\lambda - 1/b)}} \right) \mathsf{OPT},$$

which proves robustness. To prove consistency, we rewrite the RHS as follows.

$$\mathbb{E}[\mathsf{ALG}] \leq \frac{\ell}{1 - e^{-\ell/b}} \;\leq\; \frac{\ell}{1 - e^{-1/\lambda}} \;=\; \frac{1}{1 - e^{-1/\lambda}} (b + \ell - b)$$

$$\leq \frac{1}{1 - e^{-1/\lambda}} (\mathsf{OPT} + \eta) \;\leq\; \left( \frac{\lambda}{1 - e^{-\lambda}} \right) (\mathsf{OPT} + \eta). \qquad \square$$

Algorithms 2 and 3 both yield a smooth trade-off between the robustness and consistency guarantees for the ski rental problem. As shown in Figure 1, the randomized algorithm offers a much better trade-off by always guaranteeing smaller consistency for a given robustness guarantee. We remark that setting $\lambda = 1$ in Algorithms 2 and 3 allows us to recover the best deterministic and randomized algorithms for the classical ski rental problem without using predictions.

## 2.4 Extensions

Consider a generalization of the ski rental problem where we have a varying demand $x_i$ for computing resources on each day $i$. Such a situation models the problem faced while designing small enterprise data centers. System designers have the choice of buying machines at a high setup cost or renting

machines from a cloud service provider to handle the computing needs of the enterprise. One can satisfy the demand in two ways: either pay $1$ to rent one machine and satisfy one unit of demand for one day, or pay $b$ to buy a machine and use it to satisfy one unit of demand for all future days. It is easy to cast the classical ski rental problem in this framework by setting $x_i = 1$ for the first $x$ days and to 0 later. Kodialam [15] considers this generalization and gives a deterministic algorithm with a competitive ratio of 2 as well as a randomized algorithm with competitive ratio of $\frac{e}{e-1}$.

Now suppose we have predictions $y_i$ for the demand on day $i$. We define $\eta = \sum_i |x_i - y_i|$ to be the total $L_1$ error of the predictions. Both Algorithms 2 and 3 extend naturally to this setting to yield the same robustness and consistency guarantees as in Theorems 2.2 and 2.3. Our results follow from viewing an instance of ski rental with varying demand problem as $k$ disjoint instances of the classical ski rental problem, where $k$ is an upper bound on the maximum demand on any day. The proofs are similar to those in Sections 2.2 and 2.3; we omit them for brevity.

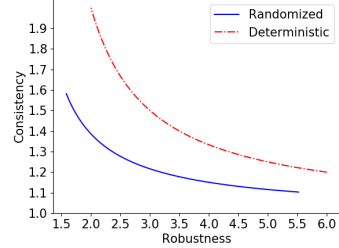

Figure 1: Ski rental: Robustness vs. consistency.

# 3 Non-clairvoyant job scheduling with prediction

We consider the simplest variant of non-clairvoyant job scheduling, i.e., scheduling $n$ jobs on a single machine with no release dates. The processing requirement $x_j$ of a job $j$ is unknown to the algorithm and only becomes known once the job has finished processing. Any job can be preempted at any time and resumed at a later time without any cost. The objective function is to minimize the sum of completion times of the jobs. Note that no algorithm can yield any non-trivial guarantees if preemptions are not allowed.

Let $x_1, \ldots, x_n$ denote the actual processing times of the $n$ jobs, which are unknown to the non-clairvoyant algorithm. In the clairvoyant case, when processing times are known up front, the optimal algorithm is to simply schedule the jobs in non-decreasing order of job lengths, i.e., shortest job first. A deterministic non-clairvoyant algorithm called *round-robin* (RR) yields a competitive ratio of 2 [23], which is known to be best possible.

Now, suppose that instead of being truly non-clairvoyant, the algorithm has an oracle that predicts the processing time of each job. Let $y_1, \ldots, y_n$ be the predicted processing times of the $n$ jobs. Then $\eta_j = |x_j - y_j|$ is the prediction error for job $j$, and $\eta = \sum_{j=1}^n \eta_j$ is the total error. We assume that there are no zero-length jobs and that units are normalized such that the actual processing time of the shortest job is at least one. Our goal in this section is to design algorithms that are both robust and consistent, i.e., can use good predictions to beat the lower bound of 2, while at the same time guaranteeing a worst-case constant competitive ratio.

## 3.1 A preferential round-robin algorithm

In scheduling problems with preemption, we can simplify exposition by talking about several jobs running concurrently on the machine, with rates that sum to at most 1. For example, in the round-robin algorithm, at any point of time, all $k$ unfinished jobs run on the machine at equal rates of $1/k$. This is just a shorthand terminology for saying that in any infinitesimal time interval, $1/k$ fraction of that interval is dedicated to running each of the jobs.

We call a non-clairvoyant scheduling algorithm *monotonic* if it has the following property: given two instances with identical inputs and actual job processing times $(x_1, \ldots, x_n)$ and $(x'_1, \ldots, x'_n)$ such that $x_j \leq x'_j$ for all $j$, the objective function value found by the algorithm for the first instance is no higher than that for the second. It is easy to see that the round-robin algorithm is monotonic.

We consider the *Shortest Predicted Job First* (SPJF) algorithm, which sorts the jobs in the increasing order of their *predicted* processing times $y_j$ and executes them to completion in that order. Note that SPJF is monotonic, because if processing times $x_j$ became smaller (with predictions $y_j$ staying the same), all jobs would finish only sooner, thus decreasing the total completion time objective. SPJF produces the optimal schedule in the case that the predictions are perfect, but for bad predictions, its worst-case performance is not bounded by a constant. To get the best of both worlds, i.e. good

performance for good predictions as well as a constant-factor approximation in the worst-case, we combine SPJF with RR using the following, calling the algorithm *Preferential Round-Robin (PRR)*.

**Lemma 3.1.** *Given two monotonic algorithms with competitive ratios $\alpha$ and $\beta$ for the minimum total completion time problem with preemptions, and a parameter $\lambda \in (0,1)$, one can obtain an algorithm with competitive ratio $\min\{\frac{\alpha}{\lambda}, \frac{\beta}{1-\lambda}\}$.*

*Proof.* The combined algorithm runs the two given algorithms in parallel. The $\alpha$-approximation (call it $A$) is run at a rate of $\lambda$, and the $\beta$-approximation ($B$) at a rate of $1 - \lambda$. Compared to running at rate 1, if algorithm $A$ runs at a slower rate of $\lambda$, all completion times increase by a factor of $1/\lambda$, so it becomes a $\frac{\alpha}{\lambda}$-approximation. Now, the fact that some of the jobs are concurrently being executed by algorithm $B$ only decreases their processing times from the point of view of $A$, so by monotonicity, this does not make the objective of $A$ any worse. Similarly, when algorithm $B$ runs at a lower rate of $1 - \lambda$, it becomes a $\frac{\beta}{1-\lambda}$-approximation, and by monotonicity can only get better from concurrency with $A$. Thus, both bounds hold simultaneously, and the overall guarantee is their minimum. $\qquad\square$

We next analyze the performance of SPJF.

**Lemma 3.2.** *The SPJF algorithm has competitive ratio at most $\left(1 + \frac{2\eta}{n}\right)$.*

*Proof.* Assume w.l.o.g. that jobs are numbered in non-decreasing order of their actual processing times, i.e. $x_1 \leq \ldots \leq x_n$. For any pair of jobs $(i,j)$, define $d(i,j)$ as the amount of job $i$ that has been executed before the completion time of job $j$. In other words, $d(i,j)$ is the amount of time by which $i$ delays $j$. Let ALG denote the output of SPJF. Then

$$\mathsf{ALG} = \sum_{j=1}^{n} x_j + \sum_{(i,j):i<j} (d(i,j) + d(j,i)).$$

For $i < j$ such that $y_i < y_j$, the shorter job is scheduled first and hence $d(i,j) + d(j,i) = x_i + 0$, but for job pairs that are wrongly predicted, the longer job is scheduled first, so $d(i,j) + d(j,i) = 0 + x_j$. This yields

$$\mathsf{ALG} = \sum_{j=1}^{n} x_j + \sum_{\substack{(i,j):i<j \\ y_i < y_j}} x_i + \sum_{\substack{(i,j):i<j \\ y_i \geq y_j}} x_j \;=\; \sum_{j=1}^{n} x_j + \sum_{(i,j):i<j} x_i + \sum_{\substack{(i,j):i<j \\ y_i \geq y_j}} (x_j - x_i)$$

$$\leq \sum_{j=1}^{n} x_j + \sum_{(i,j):i<j} x_i + \sum_{\substack{(i,j):i<j \\ y_i \geq y_j}} \eta_i + \eta_j = \mathsf{OPT} + \sum_{\substack{(i,j):i<j \\ y_i \geq y_j}} \eta_i + \eta_j \leq \mathsf{OPT} + (n-1)\eta,$$

which yields $\frac{\mathsf{ALG}}{\mathsf{OPT}} \leq 1 + \frac{(n-1)\eta}{\mathsf{OPT}}$. Now, using our assumption that all jobs have length at least 1, we have $\mathsf{OPT} \geq \frac{n(n+1)}{2}$. This yields an upper bound of $1 + \frac{2(n-1)\eta}{n(n+1)} < 1 + \frac{2\eta}{n}$ on the competitive ratio of SPJF. $\qquad\square$

We give an example showing that this bound is asymptotically tight. Suppose that there are $n - 1$ jobs with processing times 1 and one job with processing time $1 + \epsilon$ and suppose the predicted lengths are $y_j = 1$ for all jobs. Then $\eta = \epsilon$, $\mathsf{OPT} = \frac{n(n+1)}{2} + \epsilon$, and, if SPJF happens to schedule the longest job first, increasing the completion time of $n - 1$ jobs by $\epsilon$ each, $\mathsf{ALG} = \mathsf{OPT} + (n-1)\epsilon$. This gives the ratio of $\frac{\mathsf{ALG}}{\mathsf{OPT}} = 1 + \frac{2(n-1)\eta}{n(n+1)+2\epsilon}$, which approaches the bound in Lemma 3.2 as $n$ increases and $\epsilon$ decreases.

Finally, we bound the performance of the preferential round-robin algorithm.

**Theorem 3.3.** *The preferential round-robin algorithm with parameter $\lambda \in (0,1)$ has competitive ratio at most $\min\{\frac{1}{\lambda}(1 + \frac{2\eta}{n}), \frac{2}{1-\lambda}\}$. In particular, it is $\frac{2}{1-\lambda}$-robust and $\frac{1}{\lambda}$-consistent.*

*Proof.* This follows from the competitive ratio of SPJF (Lemma 3.2) and the competitive ratio of 2 for round-robin, and by combining the two algorithms using Lemma 3.1. $\qquad\square$

Setting $\lambda > 0.5$ gives an algorithm that beats the round-robin ratio of 2 in the case of sufficiently good predictions. For the special case of zero prediction errors (or, more generally, if the order of jobs sorted by $y_j$ is the same as that sorted by $x_j$), we can obtain an improved competitive ratio of $\frac{1+\lambda}{2\lambda}$ via a more sophisticated analysis.

**Theorem 3.4.** *The preferential round-robin algorithm with parameter $\lambda \in (0,1)$ has competitive ratio at most $\left(\frac{1+\lambda}{2\lambda}\right)$ when $\eta = 0$.*

*Proof.* Suppose w.l.o.g. that the jobs are sorted in non-decreasing job lengths (both actual and predicted), i.e. $x_1 \leq \cdots \leq x_n$ and $y_1 \leq \cdots \leq y_n$. Since the optimal solution schedules the jobs sequentially, we have

$$\mathsf{OPT} = \sum_{j=1}^{n}(n-j+1)x_j = \sum_{j=1}^{n}x_j + \sum_{(i,j):i<j} x_i. \tag{1}$$

We call a job *active* if it has not completed yet. When there are $k$ active jobs, the preferential round-robin algorithm executes all active jobs at a rate of $\frac{1-\lambda}{k}$, and the active job with the shortest predicted processing time (we call this job *current*) at an additional rate of $\lambda$. Note that each job $j$ finishes while being the current job. This can be shown inductively: suppose job $j-1$ finishes at time $t$. Then by time $t$, job $j$ has received strictly less processing than $j-1$, but its size is at least as big. So it has some processing remaining, which means that it becomes current at time $t$ and stays current until completion. Let *phase* $k$ of the algorithm denote the interval of time when job $k$ is current.

For any pair of jobs $(i,j)$, define $d(i,j)$ as the amount of job $i$ that has been executed before the completion time of $j$. In other words, $d(i,j)$ is the amount of time by which $i$ delays $j$. We can now express the cost of our algorithm as

$$\mathsf{ALG} = \sum_{j=1}^{n}x_j + \sum_{(i,j):i<j} d(i,j) + \sum_{(i,j):i<j} d(j,i). \tag{2}$$

If $i < j$, as job $i$ completes before job $j$, we have $d(i,j) = x_i$. To compute the last term in (2), consider any phase $k$, and let $t_k$ denote its length. In this phase, the current job $k$ executes at a rate of at least $\lambda$, which implies that $t_k \leq \frac{x_k}{\lambda}$. During phase $k$, jobs $\{k+1,...,n\}$ receive $\frac{t_k(1-\lambda)}{n-k+1}$ amount of processing each. Such a job $k+i$ delays $i$ jobs with smaller indices, namely $\{k,...,k+i-1\}$. Let $d^k(i,j)$ denote the delay in phase $k$:

$$\sum_{(i,j):i<j} d^k(j,i) = \frac{t_k(1-\lambda)}{n-k+1} \cdot \sum_{i=1}^{n-k} i = \frac{t_k(1-\lambda)(n-k)}{2} \leq \frac{x_k(1-\lambda)(n-k)}{2\lambda}.$$

Substituting back into Equation (2),

$$\mathsf{ALG} = \sum_{j=1}^{n}x_j + \sum_{(i,j):i<j} d(i,j) + \sum_{(i,j):i<j}\sum_{k=1}^{n} d^k(j,i)$$

$$\leq \sum_{j=1}^{n}x_j + \sum_{(i,j):i<j} x_i + \sum_{k=1}^{n} \frac{x_k(1-\lambda)(n-k)}{2\lambda} = \mathsf{OPT} + \sum_{k=1}^{n}\frac{x_k(1-\lambda)(n-k)}{2\lambda}$$

$$\leq \mathsf{OPT} + \frac{1-\lambda}{2\lambda}\sum_{k=1}^{n}x_k(n-k+1) = \mathsf{OPT} + \frac{1-\lambda}{2\lambda}\mathsf{OPT} = \frac{1+\lambda}{2\lambda}\mathsf{OPT},$$

using Equation (1) for the last line. $\qquad\square$

## 4 Experimental results

### 4.1 Ski rental

We test the performance of our algorithms for the ski rental problem via simulations. For all experiments, we set the cost of buying to $b = 100$ and the actual number of skiing days $x$ is a

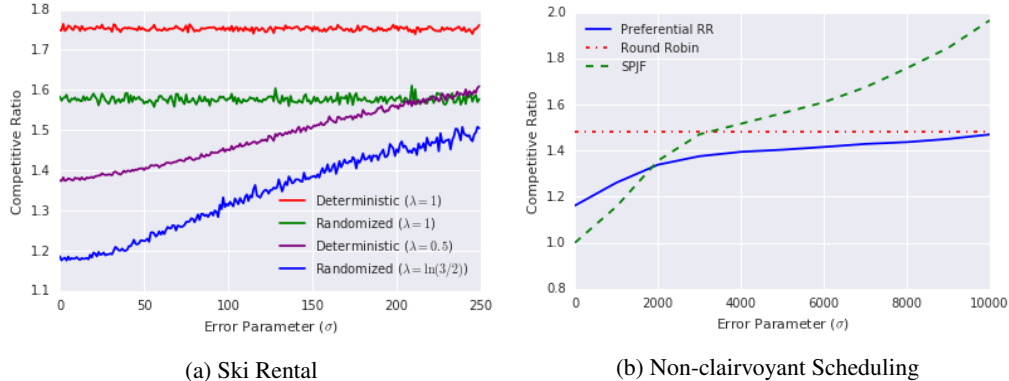

(a) Ski Rental          (b) Non-clairvoyant Scheduling

Figure 2: Average competitive ratio over varying prediction errors.

uniformly drawn integer from $[1, 4b]$. The predicted number of days $y$ is simulated as $y = x + \epsilon$ where $\epsilon$ is drawn from a normal distribution with mean 0 and standard deviation $\sigma$. We consider both randomized and deterministic algorithms for two different values of the trade-off parameter $\lambda$. Recall that by setting $\lambda = 1$, our algorithms ignore the predictions and reduce to the known optimal algorithms (deterministic and randomized, respectively) [12]. We set $\lambda = 0.5$ for the deterministic algorithm that guarantees a worst-case competitive ratio of 3. In order to obtain the same worst-case competitive ratio, we set $\lambda = \ln(3/2)$ for the randomized algorithm. For each $\sigma$, we plot the average competitive ratio obtained by each algorithm over 10000 independent trials in Figure 2a. We observe that even for rather large prediction errors, our algorithms perform significantly better than their classical counterparts. In particular, even our deterministic algorithm that uses the predictions performs better than the classical randomized algorithm for errors up to a standard deviation of $2b$.

## 4.2 Non-clairvoyant scheduling

We generate a synthetic dataset with 50 jobs where the processing time of each job is sampled independently from a Pareto distribution with an exponent of $\alpha = 1.1$. (As observed in prior work [7, 8, 2], job size distributions in a number of settings are well-modeled by a Pareto

| N | min | max | mean | $\sigma$ |
|---|---|---|---|---|
| 50 | 1 | 22352 | 2168 | 5475.42 |

Table 1: Statistics of job lengths.

distribution with $\alpha$ close to 1.) Pertinent characteristics of the generated dataset are presented in Table 1. In order to simulate predicted job lengths and compare the performance of the different algorithms with respect to the errors in the prediction, we set the predicted job length $y_i = x_i + \epsilon_i$, where $\epsilon_i$ is drawn from a normal distribution with mean zero and standard deviation $\sigma$.

Figure 2b shows the competitive ratio of the three algorithms versus varying prediction errors. For a parameter $\sigma$, we plot the average competitive ratio over 1000 independent trials where the prediction error has the specified standard deviation. As expected, the naïve strategy of scheduling jobs in non-decreasing order of their predicted job lengths (SPJF) performs very well when the errors are low, but quickly deteriorates as the errors increase. In contrast, our preferential round-robin algorithm (with $\lambda = 0.5$) performs no worse than round-robin even when the predictions have very large error.

## 5 Conclusions

In this paper we furthered the study of using ML predictions to provably improve the worst-case performance of online algorithms. There are many other important online algorithms including $k$-server, portfolio optimization, etc, and it will be interesting to see if predictions can be useful for them as well. Another research direction would be to use the error distribution of the ML predictor to further improve the bounds.

## Footnotes

[1]Informally, competitive ratio compares the worst-case performance of an online algorithm to the best offline algorithm that knows the future.

[2]The definition of the prediction error $\eta$ is problem-specific. In both the problems considered in this paper, $\eta$ is defined to be the $L_1$ norm of the error.

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
