[Reviews · NeurIPS 2018]

Reviewer 1



Main ideas ========== This paper proposes two simple and elegant algorithms for two classical problems: ski rental and single machine non-clarvoyant job scheduling. The key novelty in these algorithms is that they accept as input also a prediction algorithm that provides additional input to the problem solver. Then, the authors propose a nice analysis of the competitive ratio of the proposed algorithms as a function of the accuracy of the given predictor. For each domain (ski rental and job scheduling), the authors propose a deterministic algorithm and a randomized one, showing that the randomized algorithms perform better both in theory and in practice. Significance: The form of algorithm analysis done in this work, i.e., analyzing the robustness and consistency w.r.t prediction error, is in my view very important and under-studied. It fits well with the growing need for explainable and more predictable AI, and so I believe the significance of this work. Contributing to the signficance is also the experiments done by the authors, showing that their proposed algorithm (even the deterministic ones) outperform current approaches. Originality: I am not aware of prior work that proposed algorithms for these problems that use predictions and analyze them without any assumption about the underlying error distribution. Clarity: the paper is very clear, self-contained, and really a joy to read. Comments for improvement: ========================= 1)Lines 64-66: Here you define \gamma-robustness and \beta-consistency,but this is based on the notions of predictor and its error: either give a quick explanation of what the error is in the two problems you deal with, or just say this will be explained later. 2) Lines 134 and 136: you use d instead of b. 3) The paper is missing a conclusion and a future work section. I encourage the authors to add them.

Reviewer 2



Update after rebuttal & discussion: I am satisfied with the authors' response, and while I agree with another reviewer that a more general setup/investigation would be nice, I believe the present work focussing on special cases/problems is of sufficient novelty and marks a good first step toward more general frameworks on the future research agenda. Thus, I uphold my vote and strongly recommend accepting this submission. -------- The present paper introduces deterministic and randomized algorithms for online ski rental and non-clairvoyant single-machine scheduling problems that incorporate (machine-learned) predictions in order to improve competitiveness in case of good predictions while maintaining robustness, i.e., avoiding performance degradation, in case of bad predictions. Both problems are intuitively amenable to make use of predictions of rental or job processing times, respectively, and the proposed algorithms show how such intuition can be exploited on-the-fly for robust and competitive online solution algorithms. Thus, the work is well-motivated and contributes practically interesting methods to well-studied and relevant problems. The paper is also generally well-written and mostly easy to follow. Therefore, I recommend accepting this submission. Although the considered online-optimization problems may be easily described, they are in fact somewhat challenging. A clear strength of the present work is fusing known approximation approaches for either problem with the assumed availability of data predictions into new online algorithms that are *provably* robust in the sense that they do not deteriorate much (w.r.t. the best-known competitive ratios – i.e., ratio between computed solution and the theoretical „offline, clairvoyant“ optimum – achievable by algorithms not exploiting predictions) in case the predictions are rather bad, but are *provably* able to improve competitiveness if the incorporated predictions are good. Some numerical experiments further support the theoretical results and demonstrate practicality of the proposed schemes. I particularly like the theoretical disconnection of the introduced data-driven algorithms and the actual origin of the predictions that are exploited. A weakness of the paper is that some parts are a bit vague or unclear, and that it contains various typos or formatting issues. I shall provide details in the following list; typos I found will be listed afterwards. 1. In the section about related work, I was wondering whether there is no related literature from the robust scheduling literature. I am not overly familiar with the area, so I cannot give concrete references, but I am aware that robust scheduling is a very large field, so I would be surprised if there were no robust online scheduling problems/algorithms that are connected to the work at hand in one way or another. Could you please shed some light on this aspect (and provide some reference(s) if appropriate)? 2. From the description in the „main results“ section of the Introduction, the precise connection between competitive ratio and robustness or consistency did not become entirely clear to me. Please provide some additional clarification on what is to be compared here – in particular, I also stumbled over why (as stated in several corresponding theorems) one ends up with competitive ratios „at most min{ robustness-ratio, consistency-ratio }“. At first glance, I had thought that if an algorithm does well for good predictions but the robustness bound is bad, that the latter should dominate. As this is apparently not the case, I ask for a brief explanation/clarification, preferably already in the introduction, as to why the minimum of the two values yields a bound on the competitive ratio. 3. In Algorithm 3, please clarify what „chosen“ means exactly. (Is it „drawn uniformly at random from the k (or l, resp.) values q_i (r_i, resp.)“ ? Or „chosen as index yielding maximal q_i/r_i-value“? Or something else?) Also, on a higher level, what makes q_i and r_i probability distributions? 4. On p.5, line 151 (multi-line inequality chain): I could not immediately verify the 3rd equation (in the middle line of the chain) – if it is based on a known formula, I do not recall it (maybe provide a reference?), otherwise please elaborate. 5. Please provide a reference for the statement that „... no algorithm can yield any non-trivial guarantees if preemptions are not allowed“ (lines 187-188, p. 6, 1st paragraph of Section 3). 6. At the very end of Section 3, you omit a proof for lack of space. If possible, please do include the proof in the revision (at least as supplementary material, but maybe it fits into the main paper after all). 7. Beginning of Section 2.2: should it not be „... must incur a competitive ratio of at most $b$.“ (not „at least“) ? The worst case would be x=1, then the ratio would be b/1=b, but in all other cases (x>1), the ratio is either b/x or eventually b/b=1 (as soon as x>b). 8. In Section 2.4, perhaps add „as in Theorems 2.2 and 2.3, respectively.“ to the sentence „Both Algorithms 2 and 3 extend naturally to this setting to yield the same robustness and consistency guarantees.“ ? Finally, here's a (sorry, quite pedantic) list of typos or formatting improvement suggestions for the authors' consideration: -- line 1: „machine-learned“ (hyphen is missing here, but used subsequently) -- l. 13: I think it should be „aimed at tackling“, not „aimed to tackle“ -- l. 18: missing comma between „Here“ and „the effort...“ -- The dollar sign (used in the context of the ski rental problem) somehow looks awkward; actually, could you not simply refrain from using a specific currency? -- Regarding preemptions, I think „resume“ is more common than „restart“ (ll. 54, 186) -- l. 76: I think it should be „used naively“, not „naively used“ -- Proof of Lemma 2.1, last bullet point: to be precise, it should be „ … x < b+x-y ... “ -- l. 119: comma is missing after „i.e.“ -- Proof of Theorem 2.2: There is a „d“ that should probably be a „b“ (ll. 134 and 136) -- Throughout the paper, you use four different ways to typeset fractions: in-line, \frac, \tfrac, and that other one with a slanted „frac-line“. I would appreciate a bit more consistency in that regard. In particular, please avoid using \frac when setting fractions in-line, as in Theorem 2.2, where it messes up the line spacing. -- l. 144 (1st sentence of Sect. 2.3): comma missing after „In this section“ -- p. 5: the second inequality chain violates the textwidth-boundary – can probably easily be fixed be linebreaking before the last equality- rather than the first inequality-sign. Also, „similar to“ should be „similarly to“ (ll. 157 and 159). -- l. 216, comma missing after „i.e.“ -- Lemma 3.2: Statement should start with a „The“ (and avoid \frac in-line, cf. earlier comment) Given the end of the proof, the statement could also be made with the less-coarse bound, or writing „less than“ instead of „at most“. -- l. 234: insert „the“ between „...define d(i,j) as“ and „amount of job“ (perhaps use „portion“ instead of „amount“, too) -- l. 238: missing comma between „predicted“ and „the longer job“ -- l. 243: suggest replacing „essentially“ by „asymptotically“ -- l. 249: comma missing after „Finally“ -- Theorem 3.3: Statement should start with a „The“. -- l. 257: comma missing after „e.g.“ -- l. 276: hyphenate „well-modeled“ -- l. 279: comma missing before „where“ -- Ref. [11]: use math-mode for e/(e-1), and TCP should be upper-case -- Ref. [14]: „P.N.“, not „PN“, for Puttaswamy's initials -- Ref.s [15, 16, 17]: Journal/Source ? Page numbers? -- Generally, the reference list is somewhat inconsistent regarding capitalization (in paper and journal titles) and abbreviations (compare, e.g., [11] Proceedings title with [21] Proceedings title).

Reviewer 3



Update after rebuttal: Score updated. Summary: The paper discusses the use of ML predictions to improve online algorithms. The idea is to ensure that the new algorithms are able to maximize the utility of the added knowledge, while degrading gracefully to match at least the performance of the counterpart without ML enhancement when predictions are badly made. The idea is simple, which is nice, but the paper contains some serious issues. Strengths: Simplicity and clarity in terms of technical presentation. Weaknesses: Inaccurate statements and ambiguous details that yearn clarification (see detailed comments below). The idea is also not original. Detailed comments: The problem definition does not need to be limited to the two problems (ski rental and non-clairvoyant job scheduling). The authors could try to define a specific class of online algorithms that solve some specific type of problems, then use the above two problems as illustrations. The numerical evaluation is limited, as the two problem setups are a bit too simple. Due to the simplicity of the algorithms designed, more sophisticated evaluation with perhaps real-life data sets would make the paper stronger. Inadequate related works: The body of works focusing on expected values (or average case) and not worst case, e.g., decision theory and stochastic optimal control, is discussed very briefly at the end of page 2 and beginning of page 3. In this case, the uncertainty is modeled with probability and decisions are made based on whether expected values are maximized. Statements that are not accurate: Page 1, line 30: ML typically assumes that the model does not change over time, so the statement that ML approaches are “naturally adaptive to evolving data characteristics” is not entirely true. Unclear and needs elaboration: “Online algorithms”: There are many kinds of online algorithms, so authors should be more specific about which type they mean upfront, in the abstract and introduction. Unclear and needs elaboration: Page 1, line 31-32: Why? Intuitively if the online algo is aware of the performance of the predictor, and takes that into account, e.g., to rely more or less depending on the performance, the result will be better, no? Page 4, paragraph 3, starting from line 133: What is ‘d’? Needs to introduce ‘d’ before use Page 4, paragraph 3, starting from line 133: Consistency is the competitive ratio in the case of perfect predictor, meaning y = x. Why do you need to consider cases when x < \lambda * b, when y >= b? Shouldn’t x also be >= b when y >=b? Section 3: Reminding what job completion time means in the context of non-clairvoyant job scheduling would make it easier for readers to understand the objective function. Page 6, line 213 “SPJF is monotonic”: This is wrong. Consider the case when n=2, and two instances (3, 7) and (4, 8). Due to bad prediction, in instance 1, SPJF could execute 2nd job first, and 1st job later, with total completion time = 7 + 10 = 17. In instance 2, suppose predictions are good, and SPJF execute the jobs in that order, yielding total completion time 4 + 12 = 16. In this case, the objective value of instance 1 is higher than that of instance 2, although instance 2 has higher execution time than instance 1. SPJF is therefore non-monotonic.